

# An overview of outdoor low-cost gas-phase air quality sensor deployments: current efforts, trends, and limitations

Kristen Okorn[1,2,3], Laura T. Iraci[1]

[1]Atmospheric Science Branch, NASA Ames Research Center, Moffett Field, CA, 94035, USA
[2]NASA Postdoctoral Program, Oak Ridge Associated Universities, Oak Ridge, TN, 37830, USA
[3]now at Bay Area Environmental Research Institute, Moffett Field, CA, 94035, USA

*Correspondence to*: Kristen Okorn (kristen.e.okorn@nasa.gov)

**Abstract.** We reviewed 60 sensor networks and 15 related efforts (sensor review papers and data accessibility projects) to
better understand the landscape of stationary low-cost gas-phase sensor networks deployed in outdoor environments
worldwide. This study is not exhaustive of every gas-phase sensor network on the globe, but rather exists to categorize types
of sensor networks by their key characteristics and explore general trends. This also exposes gaps in monitoring efforts to date,
especially regarding the availability of gas-phase measurements compared to particulate matter (PM), and geographic coverage
gaps (the global south, rural areas). We categorize ground-based networks that measure gas-phase air pollutants into two main
subsets based on their deployment type: quasi-permanent (long-term) and campaign (short to medium-term) and explore
commonplace practices, strengths, and weaknesses of stationary monitoring networks. We conclude with a summary of cross-
network unification and quality control efforts. This work aims to help scientists looking to build a sensor network explore
best practices and common pathways, and aid end users in finding low-cost sensor datasets that meet their needs.

## 1 Introduction

### 1.1 A Brief History of Low-Cost Gas Sensors: Context, Development, and Growth

In the past few decades, the emergence, growth, and solidification of low-cost air quality sensor efforts have coincided with
myriad policy decisions and social movements. In this section, we will reference those in the United States (US) as examples
of possible drivers behind the expansion of low-cost sensor use. The Clean Air Act, which serves as the US's primary set of
air quality laws, was first enacted in 1963. It calls for the monitoring of six "criteria" pollutants, which are harmful to both
human health and the environment (EPA, accessed 9/18/2023). The criteria pollutants are: particulate matter (PM), ozone ($O_3$),
nitrogen oxides (NO, $NO_2$, $NO_x$), sulfur oxides, carbon monoxide, and lead. For each of these pollutants, the current guidelines
were established between 2010 and 2015 (California Air Resources Board, accessed 9/18/2023). Following the public
awareness and federal regulations of these pollutants, general air quality monitoring efforts in the US increased, and have
remained relevant to date. In 2022, the US Environmental Protection Agency (EPA) allocated over $54 million USD to fund
143 air monitoring projects in 37 states, many of which include low-cost sensor networks, including $20 million USD in grants



for ambient air monitoring in underserved communities (Afshar-Mohajer, accessed 9/18/2023). The affordability and ease-of-use of low-cost sensors continues to answer the need for more air quality monitoring, especially in historically underserved areas.

Environmental justice (EJ) refers to the unequal exposure to pollution based on factors such as race and class (Čapek, 2014), and even the placement of monitoring stations can be inequitable  (Miranda et. al., 2011). In the United States, the National Association for the Advancement of Colored People (NAACP) is largely credited with formalizing the movement with their protests in the 1980's (Department of Energy, 2024). While a variety of other special interest groups, non-profit, and community organizations have been actively involved in democratizing air pollution monitoring (National Association of

Clean Air Agencies, accessed 9/18/2023; Coalition for Clean Air, accessed 9/18/2023), formal government recognition and action on these issues may have encouraged further monitoring efforts, including low-cost sensors. Notably, in 1992 the Environmental Equity Working Group was established by then-President George H. W. Bush, followed by Executive Order 12898 issued by then-President Bill Clinton in 1994, titled "Federal Actions to Address Environmental Justice in Minority Populations and Low-Income Populations" (Department of Energy, accessed 9/18/2023). Environmental justice and air quality

monitoring in underserved regions have remained focus areas of air sensor use to date (Oyola et. al., 2022).

The rise of sensor-based exposure monitoring follows a similar timeline. In their 1960 paper, Sherwood and Greenhalgh introduced a "personal air sampler" that used a filter to collect PM (1960). While a few other studies throughout the 1960's and 1970's began to broach the idea of personal air quality monitoring (Taylor, 1987; Ott, 1982; Morgan and Morris, 1976;

National Research Council (US) Committee on Indoor Pollutants, 1981), modern day air quality sensor technology began to emerge in the 1980's, with several studies noting the use of electrochemical (Khan et. al., 1983; Ramakrishna et. al., 1989; Weppner, 1987) and non-dispersive infrared (NDIR) gas sensors (Shibata et. al., 1987), which were new at the time. During the 1990's, other commonly used sensor technologies such as thin films (Korotchenkov et. al., 1999; Meixner et. al., 1995) and metal oxides (Papadopoulos et. al., 1996; Sberveglieri et. al., 1997; Meixner et. al., 1996) began to emerge. The early

2000's brought the first few noteworthy sensor network studies (Arnold et. al., 2002; Zampolli et. al., 2004; Mahfuz and Ahmed, 2005).

The mid aughts and early 2010's brought further advancements in sensor technology (Chang et. al., 2006), major improvements to sensor calibration methods (Piedrahita et. al., 2014; Spinelle et. al., 2013), and a huge influx of sensor applications, including

the advent of large-scale sensor networks (Postolache et. al., 2009; Ma et. al., 2008) and science-grade research projects (Mead et. al., 2013; Bales et. al., 2012; Sivaraman et. al., 2013). Government programs began endorsing and funding low-cost sensors as a supplement to traditional air quality monitoring, contributing to the burst of sensor deployments during this period. For instance, the US EPA published the first version of its "Air Sensor Guidebook" in 2014, targeted as a resource for individuals interested in low-cost sensors and sensor data (2014).



During the most recent era of low-cost air quality sensing, from the mid 2010's to present, on which this review focuses, sensor projects have continued to expand with no signs of slowing down. Each the 60 sensor networks reviewed in sections 2 and 3 originated or found widespread usage during this period, which includes the growth of low-cost sensors in university and other research groups, and the advent of large-scale sensor network companies such as PurpleAir. This era represents an exciting

time when sensor networks are continuing to grow as a solution to gaps in monitoring, and gain legitimacy as monitoring tools. Figure 1 shows the number of Google Scholar search results by the decadal ranges referenced in this section for "low-cost air quality sensors outdoor stationary". While this search term is imperfect and related but irrelevant papers may still appear, it serves as a rough proxy for the explosion of low-cost sensor studies in the 2000's that is continuing into the present day.

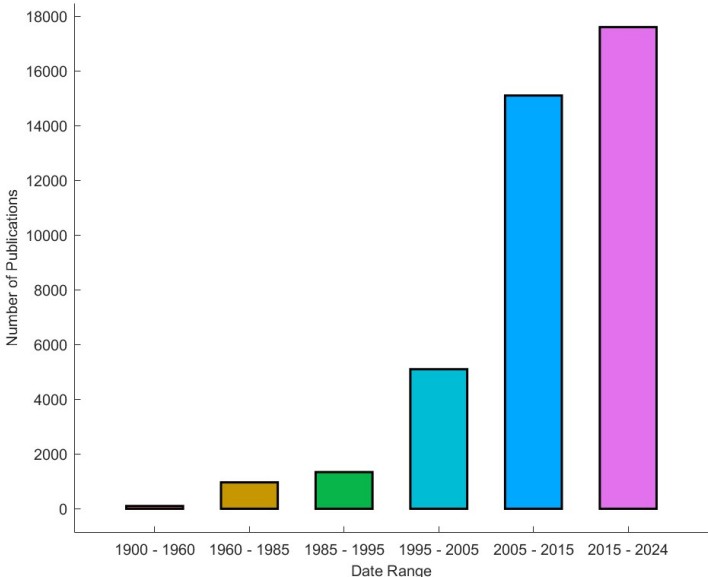

**Figure 1: Number of Google Scholar search results for "low-cost air quality sensors outdoor stationary", sorted by publication date range.**

**1.2 The Need for Gas-Phase Sensor Networks**

One of the main reasons sensor networks experienced this major growth in popularity in the twenty-first century is their ability to fill gaps in existing air quality monitoring efforts. Since low-cost sensors are a fraction of the cost of traditional air quality

monitoring instruments, a fleet of them can be deployed at the same price point as one regulatory-grade monitor. Thus, low-cost sensors have been a boon in areas that have been overlooked by traditional monitoring efforts, and alike in areas with high spatial variability of pollution where a single air quality monitor is not indicative of trends across the region.

While the findings of this study and others (Carotenuto et. al., 2023) show that PM-based sensor networks are among the

largest and most frequently deployed, there is a great need for more gas-based measurements. PM has traditionally been



targeted in monitoring efforts because of the adverse health effects associated with it (Donaldson et. al., 2005; Wyzga and Rohr, 2015; Pope and Dockery, 2006; Chen and Lippmann, 2009). PM measurements are particularly relevant to dust storms (Khader et. al., 2019) and widespread cookstove usage (Alexander et. al., 2018) in certain parts of the world, and in studying wildfires worldwide. However, even in regions where PM is the primary air quality concern, gas-phase chemistry has been
shown to drive PM emissions (Ren et. al., 2017; Quevedo et. al., 2024), necessitating data from both to understand the bigger picture.

Several volatile organic compounds (VOCs) have been shown to have severe health effects of their own, including: leukemia (Liu et. al., 2022), lung cancer (Kampa et. al., 2008), heart disease (Kampa et. al., 2008), asthma (Alford and Kumar, 2021;
Liu et. al., 2022; Ware et. al., 1993), and a variety of other acute and chronic health conditions (Fiedlier et. al., 2005; Cheng et. al., 2019; Johnston et. al., 2021). For these pollutants, both the concentration and length of exposure are crucial for understanding potential risks (Hu et. al., 2023), making it doubly important to have highly localized datasets to both assess personal exposure and set regulations relevant to health risks. Trace gases have been shown to vary widely in concentrations on opposite sides of the same building (Collier-Oxandale et. al., 2018), demonstrating that personal exposure can vary widely
even within a small radius. Again, the gas-phase data that sensors can supply on highly localized spatial and temporal scales is crucial to understanding and preventing health risks from air quality. VOCs can also contribute to ground-level ozone production, which is associated with a variety of health risks, including a variety of respiratory issues (Soares and Silva, 2022). Additionally, PM-centric sensor networks miss out on measuring greenhouse gas emissions, including methane ($CH_4$) and carbon dioxide ($CO_2$), which the present review found to be among the most studied with gas-phase sensor networks. Since
sensors are a valuable tool in filling monitoring gaps, this data can be influential for designing mitigation strategies or setting emission targets in areas lacking more widespread air pollution monitoring.

**1.3 Nomenclature**

In this work, a "sensor" refers to an individual component that typically measures one specific pollutant of interest. A "sensor package" refers to the complete measurement instrument, including one or more sensors, a circuit board, some form of data
collection, and a means of pulling the ambient sample across the sensors (e.g., a fan or pump). A "sensor network" refers to a cluster of several sensor packages deployed in some predetermined location. A "node" refers to an individual sensor package that is part of a sensor network.

**1.4 Criteria for Inclusion in the Review**

For a low-cost sensor network or associated study to be included in our review, it had to meet the following criteria:
- The cost of an individual sensor package could not exceed $10,00 USD or equivalent
- The sensor package must measure at least one gas-phase compound
- The sensor network must be ground-based



- The sensor network must be deployed outdoors
- The sensors must be deployed in a stationary location; mobile and wearable devices were excluded

We focused on application-based sensor deployments rather than reports on sensor development and calibration alone. However, in regions with few gas-phase monitoring efforts, we included deployments focusing on calibration or data quality, so long as the sensors were still deployed in a stationary outdoor location.

**1.5 Literature Review Methods**

We began our literature review with broad Google scholar searches and relevant conference proceedings, including: NASA
Health & Air Quality Applied Science Team (HAQAST) 2017 - 2023 Proceedings (NASA HAQAST, accessed 9/18/2023), European Geophysical Union (EGU) Annual Meeting 2023, Monitoring Networks Session (European Geophysical Union, accessed 9/18/2023), Air Sensors International Conference (ASIC) 2022 Proceedings (Hemingway, 2022), American Geophysical Union (AGU), Thriving Earth Exchange Projects Portal (Thriving Earth Exchange, accessed 9/18/2023), and AGU Ground-Based Atmospheric Monitoring Networks Session 2022 (AGU, accessed 9/18/2023).


Most resulting networks originated or were deployed in English-speaking or western European countries, towards which we acknowledge that our search was heavily biased, as these tend to be the areas with the most low-cost monitor coverage. For regions that were underrepresented in our initial search, we used targeted Google scholar searches to find field deployments in these areas, aiming to include multiple networks from each region. While these methods are not exhaustive, we are confident
that the trends in low-cost sensor practices have been illuminated with 60 sensor networks reviewed. This review was concluded in September 2023, with a few additional references added during the review process in early 2024.

Several air quality sensor review papers were consulted in the writing of this manuscript, which targeted key aspects of the use of low-cost sensors and directed us toward relevant papers. These included: the use of low-cost sensors for measuring
VOCs (Spinelle et. al., 2017) and particulate matter (Engel-Cox et. al., 2013), deployments in Sub-Saharan Africa (Amegah, 2018), sensor types and their calibration (Narayana et. al., 2022), two summaries of performance targets for low-cost sensors (Williams et. al., 2019; Karagulian et. al., 2019), a review of campaign-style temporary sensor networks (Carotenuto et. al., 2023), and a broad review of the use of low-cost sensors in the 2010's (Morawska et. al., 2018). Additionally, initiatives to compare low-cost sensors were utilized, including sensor comparison initiatives from: the European Network on New Sensing
Technologies for Air-Pollution Control and Environmental Sustainability (EuNetAir) (Borrego et. al., 2016), South Coast Air Quality Management District (AQMD)'s Air Quality Sensor Performance Evaluation Center (AQ-Spec) program (Collier-Oxandale et. al., 2020), and the Community Air Sensor Network (CAIRSENSE) project (Jiao et. al., 2016).



### 1.6 Format

The format of each subsequent section and subsection is as follows. First, we define the section or subsection, explaining the
characteristics of low-cost monitoring networks included within it. Following this, we dive into multiple examples of each, highlighting the characteristics mentioned in the introductory paragraph and providing more specific details and context. We conclude by summarizing the common threads in each category, exploring emerging sensor directions, and making recommendations for sensor usage and data management based on our findings.

## 2 Quasi-Permanent Low-Cost Sensor Networks

Quasi-permanent networks are deployed in stationary locations with the intent of measuring for at least two years or the remainder of the sensor's useful life. These often feature country-wide or global calibration schemes, and are the most likely to include a live dashboard for end users to visualize the data. Most sensor packages that we reviewed which fell into this category were operated by local or regional governments, or consist of widely-deployed sensor packages developed by private companies. Many of these do not list specific calibration procedures, some from a lack of standardized calibration, but others
for proprietary reasons as data is part of their product. A list of the 12 quasi-permanent networks considered in this review and their key characteristics can be found in supplemental Table S1.

### 2.1 Quasi-permanent, operated by private companies

Most sensor networks in this category are operated by for-profit companies. The typical framework is as follows: the company will manufacture and sell the sensor packages widely for profit. Each of the individually owned sensor packages becomes part
of a publicly-available data map, which anyone can visit to visualize the air quality in their area. These maps sometimes include data from regulatory monitors or other sensor networks, and typically focus on air quality index (AQI) as a key metric. While live or semi-live data is often accessible, historical data may not be available for download, or may require the use of an application programming interface (API), which may present a barrier to non-scientific users from downloading it for further inspection. Seven such companies are reviewed in this section, with the largest, PurpleAir, discussed in sect. 2.1.1.


A case study of a company born out of a university research grade project is TELLUS and its AirU sensors. The project was originally started at the University of Utah under a National Science Foundation (NSF) grant to bring higher spatial coverage air quality monitoring to the Wasatch Front region of Utah (AIRU, accessed 9/18/2023). Since then, it has evolved into a product line of indoor and outdoor sensor packages measuring CO, $NO_2$, and PM. As is typical of university research projects,
the sensors are robustly calibrated via chamber tests (Sayahi et. al., 2019; Kelly et. al., 2017), colocation with reference grade instruments (Becnel et. al., 2019a; Sayahi et. al., 2020; Kelly et. al., 2017), and the development of machine-learning based calibration algorithms (Becnel et. al., 2019b; Becnel and Gaillardon, 2021). TELLUS' website includes its own live data map dubbed AirView which includes data not only from the Tellus sensor packages, but also incorporates data from two larger



quasi-permanent monitoring networks: PurpleAir. discussed in sect. 2.1.1, and the US EPA's AirNow, discussed in sect. 2.2
(AirView Map by TELLUS, accessed 9/18/2023). This dashboard represents a recurring approach of incorporating data from
other networks to fill in gaps in spatial distribution, but does not feature a historical data archive for download. The sensors
have been utilized in at least one ender's peer-reviewed study (Mullen et. al., 2023).

IQAir, a company based in Switzerland, produces a line of low-cost sensor packages measuring $CO_2$ and PM (IQAir, aaccessed
9/18/2023). Its live data map has high spatial coverage in North America, Europe, East and South Asia, and Oceania. Their
map also includes data from third-party sensors such as PurpleAir and Clarity (IQAir Live Animated Air Quality Map, accessed
9/18/2023). Historical data and other tools can be accessed via API. Its AirVisual app uses real-time weather and global satellite
data in addition to its low-cost sensor data on the map to better predict air hourly air quality. Their website describes using
machine learning to validate and calibrate data based on the environmental conditions, and estimating the AQI where
monitoring gaps exist (IQAir AirVisual Platform, accessed 9/18/2023).

Clarity is a privately-owned company that likewise sells sensor packages meant for long-term deployments, targeting $NO_2$,
$O_3$, and PM (Clarity Movement Co., accessed 9/18/2023). It also features a live data dashboard called openmap, which includes
data from both the Clarity sensor nodes and government reference stations alike. Clarity also produces a data tool which allows
the user to view and plot historical data from individual nodes (Open Map - Clarity Movement, accessed 9/18/2023). The map
has worldwide coverage, with the highest sensor density in North America and Europe. Asia and the global south receive
considerably less spatial coverage. One end-user led deployment saw the sensors colocated with reference monitoring stations
to develop calibration algorithms (Zaidan et. al., 2020).

**2.1.1 Case Study: PurpleAir**

PurpleAir is the largest low-cost air quality sensor network across the globe, with over 22,000 PM sensors deployed across
approximately 100 countries as of 2023 (PurpleAir, Inc., accessed 9/18/23).  It offers a live global map where users can toggle
between raw sensor data and several different Air Quality Index (AQI) calculations based on the data at hand (PurpleAir Real-
Time Air Quality Map, accessed 9/18/2023). While there is no easy click-and-download option, an API can be used to access
historical data.


For quality assurance, PurpleAir relies on the sensors' dual channel laser configuration. If the two laser channels read similarly,
the data is deemed reliable. If not, the sensor will appear semi-translucent on the map with a highlighted warning if a user
selects the sensor online. The confidence score between the two channels is also displayed with either a check mark or an X
accordingly. Even with this feature, various agencies have reported that the raw PurpleAir data may exceed actual values by a
factor of two in experiments comparing them with reference-grade PM instruments (Air Quality Sensors, accessed 9/18/2023).
Likewise, the Plantower sensors used in the PurpleAir nodes have a functional life of approximately two years (PurpleAir



Community, accessed 9/18/2023), but as of 2023 there is no mechanism to remove sensor data from the map based on sensor age alone.

In recent years, PurpleAir has launched an add-on VOC sensor (Bosch BME688) to its existing PM sensor packages, but they have yet to reach the same widespread use as the original PurpleAir monitors. As seen in Fig. 2, there are far fewer VOC-enabled PurpleAir sensors worldwide than their standalone PM sensors. The AQI and calibration procedures for the VOC sensor have yet to be formalized, with only the raw sensor signal available on PurpleAir's interactive map rather than a calibrated mixing ratio estimate. Even within wealthier countries, socioeconomic disparities affect the spatial distribution of PurpleAir monitors (Sun et. al., 2022).


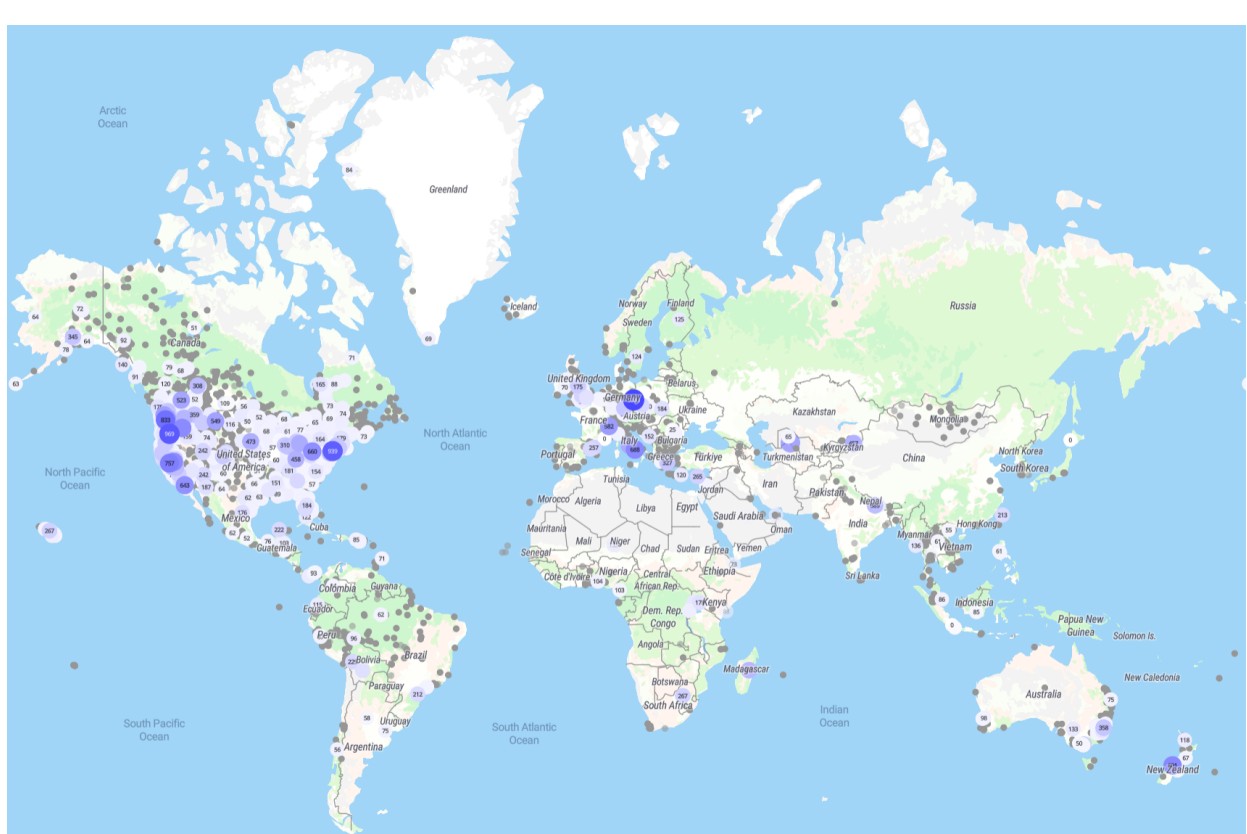

**Figure 2: Usage of outdoor PurpleAir VOC sensors (colored in shades of purple based on concentration) with additional PurpleAir PM-only sensors marked in gray. The map shows that global coverage for VOCs is sparser than for PM, with the global south and rural areas lacking gas-phase coverage. PM coverage is also significantly less dense in the global south. (Copyright PurpleAir, source:**
**map.purpleair.com, retrieved 2023/09/18).**

Since PurpleAir is an affordable, user-friendly sensor with built-in mapping capabilities, many research groups utilize their products to collect air quality data. However, since the data quality is not necessarily science-grade, many research groups have conducted pilot deployments to generate more reliable calibration algorithms, rendering the PurpleAir data more useful. We encountered many cases of research groups using the PurpleAir monitors as their low-cost sensing platform, often opting



to develop their own calibration algorithms to ensure accuracy. Studies concerning the PurpleAir VOC sensor and its
calibration are few as of 2023; although the VOC sensor is a newer addition to the PurpleAir product line, they appear to be
underutilized in addition to being under-deployed. We reviewed one study which compared the VOC sensor data to an air
quality dispersion model of VOC mixtures emanating from an industrial facility in the northeastern United States (Rosmarin
et. al., 2023). Another study in the Ohio River Valley in the US utilized both PurpleAir and AirViz monitors to measure PM

and VOCs. While regression-based calibration algorithms were derived for PM, raw VOC outputs were used for qualitative
analysis only (Raheja et. al., 2022).

Most other studies concerning PurpleAir have focused on PM sensing. Many have focused on improving calibration for
monitoring during fire and smoke events (Barkjohn et. al., 2020; Barkjohn et. al., 2022; Sablan et. al., 2022), while others have

used them in community-level monitoring and citizen science applications (Lu et. al., 2022; AirWatchSTL, accessed
9/18/2023; Jing et. al., 2022; Byrne et. al., 2023). Additional US-based studies have utilized the PurpleAir monitors and
developed their own calibration algorithms for general usage (Malings et. al., 2020; Tryner et. al., 2020; Wallace et. al., 2022).
Internationally, PurpleAir PM data has also been utilized extensively, especially in the global south where gas-phase
measurements and air quality measurements generally are sparse compared to other regions (BKH Outdoor Sensor Network,

accessed 9/18/2024; Awokola et. al., 2020; Awokola et. al., 2022; Velasquez et. al., 2022; Coker et. al., 2022; Campmier et.
al., 2023).

A host of other air quality monitoring tools incorporate data from PurpleAir into their interface, including the US EPA's
AirNow's Fire & Smoke Map (AirNow Fire and Smoke Map accessed 9/18/2023), AirView by Tellus (AirView Map by

TELLUS, accessed 9/18/2023), AirVisual by IQAir (IQAir Live Animated Air Quality Map, accessed 9/18/2023), Air Matters
(mobile app, accessed 9/18/2023), Windy (mobile app, accessed 9/18/2023), Paku for PurpleAir (mobile app, accessed
9/18/2023), Local Haze (mobile app, accessed 9/18/2023), and AirCare (mobile app, accessed 9/18/2023).

## 2.2 Quasi-permanent, operated by government

A small number of quasi-permanent low-cost sensor networks operated by various government entities were reviewed to

understand how sensors are being used to complement traditional regulatory tools. These government-operated networks
typically make use of commercially available sensors (although not always), include a live data dashboard for easy data
visualization, and incorporate regulatory monitoring station data for comparison on the data dashboard or elsewhere on their
site.

The most widely recognized usage of low-cost air quality sensors by the US government is the Environmental Protection
Agency's (EPA) AirNow program. While the main AirNow interactive map only displays regulatory-grade ozone and PM
data from government agencies (AirNow, accessed 9/18/2023), the AirNow Fire & Smoke map also includes data from



PurpleAir's low-cost PM sensors. The inclusion of these sensors greatly improves spatial coverage, but reduces confidence in the data as the PurpleAir sensors are not held to the same rigorous standards as the government-run analyzers. However, the

Fire & Smoke map is targeted towards the public for real-time decision-making during wildfire and other emission events, with the locations of fires also shown on the map and resources about the fires and safety available (AirNow Fire and Smoke Map: Getting Started, accessed 9/18/2023). Data from the main AirNow site has been used in scientific applications (Chai et. al., 2017) in addition to individual and community-level decision making.

Several municipalities and programs in the United States use Lunar Outpost's Canary sensor packages for local and regional monitoring, including the US Forest Service, US EPA, and the Denver Department of Public Health and Environment's Love my Air program (Outpost Environmental, accessed 3/8/2023). Love my Air has measured PM at schools in the Denver metropolitan area using the Canary modules since 2018, and now includes a few sites measuring a broader range of pollutants, including: $O_3$, CO, NO, $NO_2$, $H_2S$, $SO_2$, and VOCs. It includes its own data visualization map with data from both the sensors

(marked with circles) and nearby regulatory monitors (notated with squares) (Love My Air, accessed 9/18/2023). The program aims to raise community awareness about local air quality issues, and includes a downloadable education outreach curriculum for teachers to share with their students. In a similar vein, the Love my Air Wisconsin program began in the fall of 2023 (Children's Health Alliance of Wisconsin, accessed 3/8/2024).

Government-run air quality monitoring utilizing low-cost sensors is most prevalent in the US and western Europe. The Dutch National Institute for Public Health and the Environment (RIVM) has incorporated low-cost sensors measuring $NO_2$ and PM into its monitoring and citizen science projects, with approximately 30 sensors being operated directly by RIVM as of 2019. Outdoor colocation with a reference measurement and multivariate linear regression are used to calibrate the $NO_2$ sensors, and the humidity dependence of the PM sensors has also been studied (Wesseling et. al., 2019). RIVM has also created an

interactive map ("Measuring Together") to merge data from a variety of sensors - both their own and those deployed by various citizen scientists - with other public information including regulatory-grade air quality measurements, real-time traffic, and satellite data (RIVM Data Portal, accessed 9/18/2023). For individuals to add their data to the site, they must include information on the type of sensor and how it was calibrated to ensure data quality. As of 2019, RIVM has been working on an API to allow end users to download data from the site in bulk (Wesseling et. al.).

**2.3 Quasi-permanent, operated by research institution or university**

Few long-term monitoring networks are operated by research institutions or universities; most networks in this category are utilized for short-term deployments (see sect. 3.1) or as commercial products (e.g Tellus, described in sect. 2.1). However, one notable sensor network in this category is the Berkeley Environmental Air-quality & $CO_2$ Network (BEACO2N) (BEACO2N, accessed 4/8/2023). The BEACO2N nodes are widely deployed in the San Francisco Bay Area, with a few nodes in other select

cities worldwide. BEACO2N focuses on $CO_2$ measurements, but also includes sensors for a variety of other pollutants ($O_3$,



NO$_2$, CO, PM), making them broadly applicable in various monitoring efforts. Their website features both a live map of sensor readings and an easy-to-use data selection tool, which allows users to download historical data for selected date and location ranges. Their data also features robust calibrations based on comparison with reference-grade instruments (Shusterman et. al., 2016) and field calibration and evaluation (Kim et. al., 2018). While the global coverage of this network is not as dense as

many of the commercialized networks discussed in sect. 2.1, BEACO$_2$N's data quality is high due to their data management practices, as compared to others which leave quality assurance up to the end users.

Another project with long-term goals is SmartCitizen, developed by Fab Lab Barcelona in Spain. Offerings include two different varieties of sensor packages: a build-it-yourself kit, targeted at citizen scientists and educators, and a more robust

sensor platform designed for long-term outdoor use, with more flexibility of sensors to be included and higher accuracy. (Smart Citizen Kit, accessed 9/18/2023).  The latter can accommodate sensors for CO, CO$_2$, NO$_2$, O$_3$, VOCs, and PM (Camprodon et. al., 2019). It also features a global map with sensors deployed on each continent, which is still live as of 2024, although few nodes still appear to be logging live data (Map of SmartCitizen Kits, accessed 9/18/2023). While the environmental dependencies of the sensors and field validation efforts are offered, there is no explicit scheme to correct the sensor readings,

instead utilizing the raw signals from each (Camprodon et. al., 2019).

### 2.4 Quasi-permanent, operated by non-profit

Here we discuss two different models of non-profit sensor networks in the quasi-permanent category: sub-networks that employ monitors purchased from the large, privatized sensor manufacturers, and non-profit sensor networks that build out their own sensor packages. Most serve a single community or region facing air quality issues. These networks tend to exist on

smaller scales than their for-profit counterparts, with a smaller number of nodes per network and smaller spatial reach. While each of the networks we encountered in this category featured a live data map, we did not encounter any with a historical data archive or download option, nor any obvious explanation of data quality or calibration procedures on their websites. However, even if the data is not science-grade, these networks can be crucial to building community interest and understanding of local air quality issues.


One example of a non-profit utilizing commercially purchased sensors is MKE Fresh Air Collective (MKE Fresh Air Collective, accessed 9/18/2023), a community-led non-profit organization based in Milwaukee, Wisconsin, USA. It aims to provide neighborhood-scale monitoring and address environmental racism through community action. It utilizes sensors purchased via IQAir, and the data can be viewed by zooming to Milwaukee on the main IQAir map. This is an attractive model

for community groups and other organizations or individuals with low technical skills and/or limited time to utilize pre-existing resources rather than having to develop their own. However, this also often means that the non-profit is limited to the calibration and visualization tools of the host network, which may be inadequate for some user needs.



A small number of non-profit organizations develop their own sensor packages and make them available for sale, typically on
a much smaller scale than the for-profit networks. One example is Ribbit Network (Ribbit Network, accessed 9/18/2023), a
non-profit organization that targets teachers and citizen scientists to deploy their $CO_2$ sensors. Their offerings include a pre-
fabricated sensor package, or the materials for the end user to build it themselves, which is targeted towards K-12 education
outreach efforts. Ribbit likewise has developed its own global map to view $CO_2$ levels from its sensors worldwide, and offers
publicly available short-term data downloads. While calibrations are underway, there are no peer-reviewed studies using this
network as of early 2024.

## 3 Campaign-Based Low-Cost Sensor Networks

The remainder of the sensor networks reviewed are campaign-based, deploying sensors for a limited timeframe to study a point
source or phenomenon local to the geographic area. The data quality in this category varies widely, with government and
university groups typically producing the highest quality data, while non-profit or smaller commercial networks may include
less quality assurance. A few examples from each subcategory are described in detail here, with at least one example from a
country outside the US or western Europe for each. Nearly 50 campaign-based networks are summarized here, with myriad
others that sit slightly outside the scope of our review cited for further reading, but not discussed in detail. All of the former
can be found in supplemental Table S2.

### 3.1 Campaign, operated by research institution or university

We encountered more research-grade campaign-based studies using low-cost air quality sensors than all other categories
combined. In this section, we select a few key projects from each global region to review in detail. The US, western Europe,
and parts of southeast Asia are more thoroughly covered in these studies than any other region, and a portion of studies
concerning the global south were conducted by American or western European institutions, albeit usually with local
collaborators. Research-grade projects tend to have the highest data quality and most robust calibration procedures, including
an ambient colocation with a reference grade instrument and an algorithm to relate the signals of each. These also tend to
include the highest number of pollutants being measured by each sensor package. However, these often lack a central data
portal for stakeholders to visualize or access data easily. Note that at least one example deployment is included for each
campaign, but is not intended to be an exhaustive list of every use case of each sensor network. Regional boundaries have been
defined for illustrative purposes only; regions do not necessarily represent similarly-sized populations, nor an equal number
of countries. While most of the sensor packages utilized in this category are in-house built by the research institution, we
encountered a small number of research institutions that made use of sensor packages built by commercial manufacturers. In
these cases, we list the name of the manufacturer, and also cite the research group responsible for deploying them.



### 3.1.1 Campaign, operated by research institution or university in North America

Campaign-based low-cost gas-phase sensor networks are abundant in North America, especially the United States. In Canada, we found that PM studies were more common, including the measurement of wildfire smoke (Si et. al., 2020; Lee et. al., 2024; Jaafar et. al., 2024). A research team at Carnegie Mellon University developed the Real-time Affordable Multi-Pollutant (RAMP) air quality sensor packages, measuring CO, $NO_2$, $O_3$, $SO_2$, and PM. A dense sensor network was established near Pittsburgh, Pennsylvania, USA, to better understand urban emissions (Tanzer et. al., 2019; Li et. al., 2019; Tanzer-Gruener et.

al., 2020), and a shorter-term campaign was conducted in San Juan, Puerto Rico following Hurricane Maria in 2017 (Subramanian et. al., 2018). Another set of sensors was deployed in British Columbia, Canada, to characterize emissions inside of a parking garage (Liu et. al., 2021). These sensors have also been deployed in various African countries (Bahino et. al., 2021). Calibration efforts include colocation with a reference instrument and subsequent modeling including: linear and quadratic regression, clustering, neural networks, Gaussian processes, and hybrid random forest models (Malings et. al., 2019;

Zimmerman et. al., 2018; Jain et. al., 2021). Their data is hosted on Google Drive for greater public access.

Another noteworthy US-based project is the University of Colorado Boulder's "pods". The in-house built nodes contain sensors for: CO, $CO_2$, $NO_2$, $O_3$, VOCs, and PM (Piedrahita et. al., 2014). Several studies detail the calibration and validation of their sensors, which typically utilize an outdoor colocation with a reference grade instrument and the use of machine learning

techniques such as artificial neural networks (Casey et. al., 2019) and multivariate linear regression (Collier-Oxandale et. al., 2018). Additional techniques have rendered its VOC sensors applicable for speciated hydrocarbons such as methane and formaldehyde (Okorn and Hannigan, 2021) in oil and gas monitoring applications (Cheadle et. al., 2017; Collier-Oxandale et. al., 2020; Okorn et. al., 2021). An offshoot of the network, Inexpensive Network Sensor Technology for Exploring Pollution (INSTEP), uses the same sensors in applications primarily concerning comparison with remote sensing instruments (NASA

Ames INSTEP, accessed 3/8/2024). Neither iteration of the project features a public-facing data dashboard or download option.

While the ARISense low-cost monitors for CO, NO, $NO_2$, $O_3$, and PM are primarily US-based, they have also been deployed in Malawi in East Africa. The sensor packages were built in-house by Cross et. al. (2017), and have since been used by researchers with a variety of affiliations, including North Carolina State University. After colocation with a reference-grade

instrument, a series of different machine learning models were tested to best fit the data: hybrid k-nearest neighbors, random forest hybrid, high-dimensional model representation, quadratic regression, and multivariate linear regression (Bittner et. al., 2022). Deployments in both Massachusetts, USA and Malawi were effective (Cross et. al. 2017), although the authors note the lack of reference monitoring stations in Malawi hindered their ability to regularly re-calibrate the sensor nodes.



### 3.1.2 Campaign, operated by research institution or university in Europe

Spatial coverage of low-cost sensors in Europe is among the highest of all the regions, but this is dominated by sensor usage in western European cities; rural areas and eastern Europe lack coverage. A recent ozone sensor calibration study in Poland (Badura et. al., 2022) suggests that outdoor sensor usage is beginning to gain traction, but we only encountered a few application-based studies in eastern Europe.

One eastern European example is the Czech Technical University in Prague's Trafficsensnet project. It involved deploying approximately 20 in-house built sensor packages measuring CO, $NO_2$, $SO_2$, VOCs, and PM in Prague to better understand the air quality effects of traffic in the urban area. Colocation with reference grade instruments was used to validate the sensor readings, although no additional calibration algorithms were fit (Brynda et. al., 2015). The project was later expanded to include an online data portal and map-style viewer (Brynda et. al, 2016), which is unusual for university research grade
projects. In later iterations of the project, however, the gas-phase sensors were removed to focus on PM (Brynda et. al. 2020).

We also include studies conducted in Siberia or Russia with the eastern European countries as they exhibit a similar lack of coverage, albeit for a much larger geographic area. We encountered one PM study in the region (Lin et. al., 2020) and one study using spatially-distributed passive sampling (Khuriganova et. al., 2019), but no true gas-phase low-cost sensor studies,
highlighting the need for more monitoring and availability of data in this region.

The remaining sensor networks reviewed in this section hail from western European countries, which are generally wealthier (Brzeziński et. al., 2020) and see much higher low-cost sensor usage. A recent project that served several European countries was Barcelona Technical University's in-house built CAPTOR box measuring ozone in Spain, Italy, and Austria (Barcelo-
Ordinas et. al., 2021). Outdoor colocations with a reference instrument were conducted in each country, and different linear regression models were used to calibrate each set of sensor packages. This CAPTOR deployment was one of the few that serviced urban, suburban, and rural sites in one deployment to examine differences among them (Barcelo-Ordinas et. al., 2019).

The CarboSense project out of ETH Zurich, Switzerland, deployed over 300 sensor nodes measuring $CO_2$ throughout
Switzerland, with many of them concentrated in the city of Zurich. The sensors were manufactured by Decentlab GmbH and calibrated in-lab as well as via ambient colocation with a regulatory grade instrument. The subsequent calibration algorithms were based on the Beer-Lambert law and included corrections for abnormal humidity readings and drift correction over time. The sensors showed high accuracy during the approximately two-year long deployment, but were insufficient in capturing small regional gradients ( Müller et. al., 2020).




The AIRQino platform was developed by the Italian National Research Council as a sensor package to measure CO, $CO_2$, $NO_2$, $O_3$, and PM. While it has primarily been used in mobile (Zaldei et. al., 2015) and PM-focused studies (Brilli et. al., 2020; Brilli et. al., 2021), one recent application tested its effectiveness for measuring $CO_2$ in the arctic in several locations on Svalbard, Norway, an island in the arctic circle. The sensors were colocated outdoors with a reference instrument and although 425 no further calibration algorithms were derived, the AIRQino sensors showed strong agreement with the reference signals, even in the extreme cold (Carotenuto et. al., 2020).

Several low-cost monitoring projects have sprung up in France in recent years, including the Cairbox monitoring platform, which has been used to monitor reduced sulfur compounds $H_2S$ and $CH_3SH$ at and around a wastewater treatment plant. This 430 study was a collaboration between researchers from IMT Mines Alès, a university in France, and the sensor manufactuer, Cairpol. The sensors showed strong agreement with theoretical concentrations, but no calibration factors were applied (Pereira-Rodrigues et. al., 2010). Despite the lack of further studies with this sensor platform, we have opted to include it here as it is one of the few sulfur compound monitoring efforts we encountered.

The UK is a hotbed for low-cost sensor studies, including some high-profile deployments. A network of sensor packages 435 known as SNAQ boxes (Sensor boxes for Air Quality measurements) designed and built by the University of Cambridge measure CO, $CO_2$, NO, $NO_2$, $O_3$, $SO_2$, VOCs, and PM. In-laboratory calibrations were conducted, and were further validated via colocation with an outdoor regulatory grade instrument (Popoola et. al, 2018). The SNAQ boxes have been utilized in applications ranging from a high-density network in and around the London Heathrow Airport (Popoola et. al, 2018) and international deployments in Nigeria (Omokungbe et. al., 2023), China (Ouyang et. al., 2016; Shi et. al., 2019), and Portugal 440 (Borrego et. al., 2016).

Elsewhere in the UK, researchers at the University of Sheffield have made use of Envirowatch Ltd.'s "E-MOTE" sensor packages to measure CO, NO, and $NO_2$ throughout the city. Note that although the complete sensor packages were purchased from a manufacturer, this study is included under university efforts because the manufacturer does not appear to provide 445 recommendations or resources for calibration, data storage, or analysis as of 2024 (E-MOTE – Envirowatch Ltd., accessed 4/8/2024). For this study, the sensor packages were calibrated by the university team via colocation with a regulatory grade instrument and subsequent supervised machine learning approaches, including linear regression and generalized additive models. Overall, the sensor network was able to demonstrate the diurnal, weekly, and annual variation of pollutants in Sheffield (Munir et. al., 2019).


### 3.1.3 Campaign, operated by research institution or university in the Middle East

While very few sensor network studies within our criteria for inclusion have been conducted in the Middle East to date, a few related studies suggest that sensor use in the region may be gaining momentum. We came across several of PM-focused studies





in the region (Khader et. al., 2019; Saleh and Kader et. al., 2022; Karaoghlanian et. al., 2022; Yavus et. al., 2021; Ozler, 2018),
and one using air quality sensors onboard an uncrewed aerial vehicle (Al-Hajjaji et. al., 2017). A recent study from the Cyprus
Institute performed a yearlong outdoor colocation of low-cost CO, $NO_2$, $O_3$ and $SO_2$ sensors with reference-grade instruments
(Papaconstantinou et. al., 2023); while this study lacked a wider field component, it marks an important step for sensor
measurements in the region.

A notable study comes from the Israel Institute of Technology, where CanarIT sensor packages were deployed to measure
$NO_2$, $O_3$, VOCs, and PM at sub-neighborhood scale. Linear regressions were used to improve data quality after the sensor
packages were colocated with reference grade instruments. A "rolling forwards field calibration procedure" was also
introduced in which calibration coefficients for each subsequent day were based on reference data from the previous three
days, but would require further testing to be considered a viable method. Overall, the deployment identified hotspots within
the neighborhood (Moltchanov et. al., 2015).

In Doha, Qatar, researchers from the Qatar Mobility Innovations Center developed a low-cost sensor platform dubbed MGMS:
Multi-Gas Monitoring Stations. Each node included in the initial study measured $O_3$, $NO_2$, $H_2S$, and CO, while additional
nodes added later measured each of these in addition to $SO_2$, NO, and PM. The team used the sensors to estimate the AQI
around various locales of the city rather than precise concentrations, so robust calibrations were not used in this aim (Yaacoub
et. al., 2013).

### 3.1.4 Campaign, operated by research institution or university in South Asia

While there are several notable monitoring efforts in India, overall monitor density is low (Brauer et. al., 2019) and largely
focuses on urban areas (Agrawal et. al., 2021). Other monitoring efforts in India focus on PM (Sahu et. al., 2020), but will not
be described in detail here as they lie outside the scope of gas-phase monitoring. Recently, a US-based research group deployed
their MPAQS (Multi-Pollutant Air Quality Sensor) packages in Delhi, India, measuring CO, $NO_2$, $O_3$, $SO_2$, and PM. Although
colocations with research-grade instruments took place, calibration algorithms were not developed in this application, and
factor analysis was later used to relate these measurements to local sources (Hagan et. al., 2019).

At Lahore University of Management Sciences in Pakistan, the VIEW (Volunteer Internet-based Environment Watch) project
aimed to monitor urban emissions. The in-house built sensor packages measured CO, $O_3$, and $SO_2$ and were deployed around
Lahore. Detailed information on their calibration and deployment results have yet to be formally released, but this project still
marks a rare low-cost monitoring network in Pakistan. Another sensor network measuring CO and PM performed its pilot
study in Vadodara, India. Preliminary results from this study further demonstrate the need for more gas-based monitoring in
the country (Barot et. al., 2020).




Elsewhere in India, the SATVAM (Streaming Analytics over Temporal Variables from Air quality Monitoring) initiative tested various calibration schemes and proposed a non-parametric algorithm as the most robust to sensors located in vastly different environments: Delhi and Mumbai. These in-house built sensor packages built by the Indian Institute of Technology measured
$NO_2$ and $O_3$. The proposed calibration algorithm consisted of a distance-weighted k nearest neighbors approach with a learned metric, which proved effective when deploying the monitors across cities and seasons (Sahu et. al., 2021).

### 3.1.5 Campaign, operated by research institution or university in East Asia

Air quality issues in East Asia are complex (Jeong et. al., 2013), and in recent years low-cost gas-phase sensors have been able
to help fill gaps in monitoring throughout the region, though most studies have concerned urban areas. Most East Asian studies we encountered were based out of China.

A research team from the Chinese Academy of Sciences used in-house built sensor packages to measure CO, $NO_2$, $O_3$, and $SO_2$ in Beijing, China. Throughout the yearlong study, the sensors were robustly calibrated using single and multiple linear
regression, random forest regression, and neural networks after outdoor colocation with nearby reference grade instruments. The sensors were tested under different pollution levels and environmental conditions, although the resulting data appeared to be biased based on the calibration model (Han et. al., 2021).

Another study in Beijing, China, was carried out by Tsinghua University and collaborators at two universities based in the US.
In-house built sensor packages were deployed on rooftops throughout Beijing to measure $CO_2$, NO, $NO_2$, $O_3$, and PM. The sensors were calibrated for ozone only via colocation with a reference instrument, and best-fit linear regressions were generated. Sizable variation was seen in outdoor ozone concentrations across the city (Liu et. al., 2020).

A research group at the City University of Hong Kong developed their own MAS (Mini Air System) to monitor ambient CO,
$NO_2$, and PM in Beijing, China. The sensors were calibrated in both laboratory and field settings. After an outdoor colocation with reference grade instruments, linear corrections were applied to the gas-phase sensor outputs. One notable application included the use of the sensor network to measure pollution at stationary locations along the Beijing Marathon route (Sun et. al., 2016).

Another study sought to characterize ozone in Hong Kong's greater bay area using Portable Air Stations (PAS) sourced from Sapiens, a private company. A team from the Hong Kong University of Science and Technology deployed the PAS monitors to measure NO, $NO_2$, and $O_3$. An outdoor calibration was conducted with a government-run air quality monitoring station prior to deployment, and the sensor measurements were later used as inputs in a chemical transport model (The Nested Air Quality Prediction Modeling System, NAQPMS) to better characterize a high ozone event that occurred in 2016 (Chen et. al., 2024).




Outside of China, a Tokyo Institute of Technology study measured $NO_2$ concentrations in the university's surrounding area using in-house built sensor platforms. An in-lab calibration was utilized to correct the sensor response for different temperature and humidity conditions. Through the sensor's wireless network, measured $NO_2$ concentrations were frequently compared with other sensors and nearby regulatory grade monitors for auto-calibration purposes. Overall, the network proved effective

in discerning air quality differences throughout the city, including anomalies in $NO_2$ (Tsujita et. al., 2005).

At the National Taiwan University, in-house built CO sensor packages were deployed throughout Taipei near major roadways to determine the air quality effects of traffic in the city. The sensors were colocated at an EPA reference station, and a segmented linear regression approach was used to calibrate the sensors, since there only appeared to be a linear relationship

between sensor voltage and reference concentration above a certain threshold. The study provided street-level insights into pollution from traffic, with tangible differences between weekdays and weekends as well as between rush hour and regular traffic (Wen et. al., 2013).

Researchers at Seoul National University employed a network of self-built $CO_2$ monitors dubbed SNUCO2M (Seoul National

University $CO_2$ Measurement) to better understand $CO_2$ fluctuations in Seoul, South Korea. The sensors were colocated with reference grade instruments to ensure reliable functioning before full deployment in Seoul. The study demonstrated the characteristics of urban $CO_2$ throughout the city (Park et. al., 2021).

**3.1.6 Campaign, operated by research institution or university in Oceania**

One uncommon instance of a campaign-based sensor network with a live data dashboard is the Knowing Our Ambient Local

Air-quality (KOALA) project at Queensland University of Technology in Australia (Queensland University of Technology, accessed 3/10/2024). KOALA offers measurements of CO and PM in a variety of locations, including deployments in Australia, China, Vietnam, Sri Lanka, and a number of islands in Oceania. Since this network is deployment based, not all regions of the map are active, and there is no historical archive for end users to download data from. However, as is typical of university-grade research projects, the KOALA monitors are calibrated both indoors and outdoors with a reference-grade

instrument to ensure proper performance (Jayaratne et. al., 2021; Liu et. al., 2020; Kuhn et. al., 2021).

Another sensor campaign in the Oceania region used Aeroqual ozone sensors in Auckland, New Zealand, conducted by the University of Auckland (Miskell et. al., 2017). The same general framework and ozone sensors were also used in a study in British Columbia, Canada. An in-lab calibration scheme was utilized to apply simple linear corrections to the ozone data (Bart

et. al., 2014) as well as an outdoor colocation with a reference-grade instrument, and field comparisons of individual sensors. Both studies aimed to establish best practices for deploying networked sensors, including the examination of intra- and inter-site differences, and the effects of local topography and meteorological conditions on sensor readings.



### 3.1.7 Campaign, operated by research institution or university in Africa

In Africa, we encountered numerous studies using low-cost sensors to monitor PM (Singh et. al., 2021; Abera et. al., 2020;
Tékouabou et. al., 2022; Pope et. al., 2018; Raheja et. al., 2022; Hodoli et. al., 2023; Lassman et. al., 2020; Giordano et. al.,
2021; Omokungbe et. al., 2020; Gryech et. al., 2020; Gnamien et. al., 2021; Ngo et. al., 2019; Bainomugisha et. al., 2023;
Ndamuzi et. al., 2023), but far fewer targeting gas phase compounds. One reason for this is the prevalence of cookstoves in
certain African countries, for which PM is a greater concern than any gaseous pollutant (Alexander et. al., 2018). The gas-
phase sensor studies we reviewed were less likely to include robust calibration techniques than those of higher income regions.

Carnegie Mellon University's RAMP sensor network, described in more detail in sect. 3.1.1, has deployed five sensors in Cote
d'lvoire and another four in Accra, Ghana, as part of their "Improving Air Quality in West Africa" project (Bahino et. al.,
2021). These ongoing studies include colocation of several sensors with reference monitors for improved calibration.

In Nairobi, a team of researchers from Massachusetts Institute of Technology and their collaborators deployed a network of
in-house built sensors measuring NO, $NO_2$, $SO_2$, and PM. This study represents the first time that air quality at schools was
studied in Nairobi, thus student and community engagement played a large role in the study. Since the gas sensors exhibited
temperature dependencies, a temperature correction was used to ensure data quality. Despite the technical limitations of the
sensors, in this context their data was sufficient to begin understanding the differences in air quality at the different schools
studied (deSouza et. al., 2017).

In Nigeria, one study deployed five low-cost sensors in a semi-urban environment to measure ambient NO, $NO_2$, CO, $CO_2$,
$O_3$, and PM. Researchers from Obafemi Awolowo University did not specify any true calibration procedures in their
manuscript, but did note previous colocations of the same brand of sensor used with reference-grade instruments in the UK by
another research group. Overall, the study was able to examine spatial differences among the five test locations (Owoade et.
al., 2021). In another Nigerian study, researchers from the Federal University of Technology Nigeria used portable CO loggers
to compare urban and rural pollution. Since a prefabricated monitor was used, no additional calibration or quality assurance
were completed (Balogun et. al., 2014).

A Moroccan team  from Abdelmalek Essaadi University created and deployed their own low-cost sensor node measuring $CO_2$,
$NO_2$, and meteorological data with the goal of creating an algorithm that estimated the air quality index (AQI) from the sensor
inputs alone. No gas sensor calibrations were conducted as exact concentrations were not required for their intended purpose.
The initial sensor node was successful in predicting the AQI, although the project does not appear to have been scaled up to
include more nodes yet (Fahim et. al., 2023).






Due to the lack of gas-phase stationary outdoor monitoring studies in Africa, here we list a few related studies outside the scope of our review for further reading. We encountered a mobile monitoring low-cost sensor study of CO, $CO_2$, and PM in Senegal (Ngom et. al., 2018). Another study used low-cost sensors measuring $CO_2$, $NO_2$, $O_3$, and PM indoors in Nigeria (Adamu et. al., 2023). Although these studies do not fit precisely into our review, they still represent lower-cost methods of
better understanding gas-phase compounds in Africa, which is severely understudied.

### 3.1.8 Campaign, operated by research institution or university in Central & South America

Similar to Africa, the extent of low-cost sensors in Central and South America is extremely limited compared to other regions. Since PM is typically a greater concern than gases in the global south (Carvalho, 2016), we encountered more sensor networks measuring PM in these areas (Tagle et. al., 2020; Jang and Jung, 2023; Lopez-Restrepo et. al., 2021; Gramsch et. al., 2020;
Gramsch et. al., 2021; Candia et. al., 2018; Connerton et. al., 2023; Dirienzo et. al., 2023; Quintero et. al., 2023) than those measuring gas-phase compounds, although overall coverage is still sparse. A few mobile sensor studies are available for further reading on gas-phase coverage in Central and South America (Guevara et. al., 2010; Alvear-Puertas et. al., 2022).

Researchers at the University of Brazil performed in-lab calibrations and applied linear regressions to ensure the quality of
their low-cost CO sensors before deploying them in the local area, including during the World Cup. Data was stored in WebGIS via ArcGIS software, allowing geographic data to be integrated with the CO data (Aguiar et. al., 2015). In Argentina, low-cost sensors measuring VOCs were deployed for agricultural monitoring by a team including several researchers from the Universidad Nacional de San Luis (Valenzuela et. al., 2018). While this preliminary study focused more on the creation of the sensor platform rather than its calibration or results, it shows promise for the growth of low-cost sensor networks in South
America in the coming years.

AQMesh sensors, which are described in more detail in sect. 3.2, were deployed by the University of Leeds and local collaborators in Nicaragua for 6 months in 2017 downwind of the active Masaya volcano to monitor $SO_2$ and PM. Outdoor colocation with a reference-grade instrument was used to calibrate the sensor signals. Some nodes were more accurate than
others in detecting volcanic emissions, but were suitable for identifying PM enhancements nonetheless (Whitty et. al., 2022). Other studies in Nicaragua and neighboring countries like Costa Rica have employed low-cost sensors onboard uncrewed aerial systems (UAS) (Xi et. al., 2016; Granados-Bolaños et. al., 2021) or other low-cost techniques (Aguilera et. al., 2020) to monitor volcanic plumes.

In Peru, sensors measuring CO, $NO_2$, $SO_2$, $H_2S$, $O_3$, $CO_2$, and PM were deployed at four outdoor locations by the Pontifical Catholic University of Peru. The research group built geographic data visualization into their interface, and compared measured concentrations among sensors as well as with a nearby government-run air quality monitoring station (David et. al., 2015).





In Cienfuegos, Cuba, a low-cost sensor measuring $NO_2$, $SO_2$, $O_3$, CO, and PM was deployed by researchers from Universidad

Central "Marta Abreu" de Las Villas and collaborators, primarily for calibration and quality assessment purposes, including two laboratory calibrations and a month-long field deployment near a reference-grade monitor. We have included this calibration-focused study here due to the lack of gas-focused studies in this region, as well as the relevant local sources surrounding the monitor placement, including an oil refinery, thermoelectric plant, charcoal production plant, and diesel-related sources (González Rivero et. al., 2023; Martinez et. al., 2023).


We also encountered two low-cost sensor network pilot studies in Ecuador by researchers at Universidad San Francisco de Quito USFQ and Universidad Técnica de Cotopaxi. One measured CO, $CO_2$, and PM in three distinct locations. A previous study by the same group conducted a chamber calibration test for the sensors (Guanochanga et. al., 2018). Researchers found that for their urban site in Quito, Ecuador, the average CO value was below the target given by the local government, but

exceeded World Health Organization guidelines (Fuertes et. al., 2015).

### 3.2 Campaign, operated by private companies

Several sensor networks operated by private companies operate using campaign-based deployments. Typically, these companies have their own portals for data visualization and/or analysis, but unlike the quasi-permanent sensor network companies, this data is only available to users who purchase a sensor and a data management plan. These companies focus on

targeted air quality solutions for their clients rather than widely available public data or citizen science. Since their full data portals are typically only available on a subscription basis, we have opted to include them in the campaign category because widespread, long-term, stationary data is not publicly available at the time of writing. The data analysis included with the subscriptions is generally more advanced than those offered by public networks, including advanced calibration procedures and real-time modeling. MODULAIR and Bettair are the only two companies reviewed that directly address the effects of age

on their sensors' performance. Peer-reviewed studies concerning the use of the sensor packages in this category tend to be short-term deployments focusing on a specific point source or local phenomenon. These companies are usually headquartered in the US or Europe.

Airly requires its users to purchase a subscription to fully utilize their sensor packages. It is dually headquartered in both

Poland and California, USA.  Their target pollutants include CO, $NO_2$, $O_3$, $SO_2$, and PM (Airly Data Platform and Monitors, accessed 9/18/2023). Airly's live map includes data from government-run monitoring stations in addition to its own sensors, notated by different shapes on the map. Historical data can be viewed on the map, but only represents the past 24 hours, and the end user needs to utilize the API to download it. The map also includes a forecast extending forward 24 hours. Airly's sensor calibration procedures are regularly performed, hence incentivizing customers to maintain their subscriptions. Sensors

are calibrated in a laboratory setting followed by an outdoor colocation with a reference-grade instrument. Routine monitoring of the sensor package once deployed is a unique aspect of Airly's model, including consideration of local environmental



conditions and relevant satellite data to ensure sensor packages are performing adequately during deployment (Airly – Quality & Certificates, accessed 9/18/2023).

The MODULAIR sensor packages by QuantAQ measure CO, $CO_2$, NO, $NO_2$, $O_3$, and PM. Similar to Airly, users need to purchase access to the QuantAQ cloud to access their data and the API. Different pricing plans are offered for "basic", "pro", and "enterprise" users, the latter two of which include an annual inspection and re-calibration of the sensor, ensuring proper functioning and accuracy of the data. The details of this calibration process are not specified on their public-facing website as of 2023, but it appears that data transparency and calibration models are available to paid subscription users. Likewise, data
visualization is available within the cloud for paid users, but has yet to be made publicly available for citizen science use (QuantAQ, accessed 9/18/2023). The MOULAIR sensors have been utilized in several peer-reviewed publications (Yang et. al., 2022; Prathibha et. al., 2020).

Another company attempting to address the limited longevity of its sensors is Barcelona-based Bettair. Rather than reeling its
devices in yearly for recalibration, the company sells sensor "cartridges" that are meant to be repurchased every two years, ensuring that the sensors are easily replaced at the end of their useful lifetime without having to purchase an entirely new node. Each sensor package measures: CO, $CO_2$, NO, $NO_2$, $O_3$, $SO_2$, $H_2S$, and PM (Bettair Cities, accessed 9/18/2023). Colocation with reference-grade monitors shows strong agreement for NO, $NO_2$, and $O_3$ (Perello, 2018). Bettair offers a software platform for its users which includes mapping and other data visualization, but the data is not made publicly available. While the Bettair
nodes can be implemented as stationary devices in future studies, to date, the only peer-reviewed studies utilizing them have adapted them as wearables (Kotzagianni et. al., 2023; Vrijheid et. al., 2021).

Kunak AIR, a company based in Spain, has deployed their sensor packages for CO, $O_3$, $NO_x$, $SO_2$ and PM in short-term campaigns globally, with notable deployments including Ethiopia, Germany, India, Belgium and Spain (Ibarrola, 2022;
Hofman et. al., 2022; Saúco et. al., 2022). By opting into a paid software package called Kunak AIR Cloud, users unlock access to air quality reports, unspecified calibration and supervision of their sensors, and data visualization and analysis tools, including mapping and basic plotting features (Kunak AIR, accessed 9/18/2023). This company shares many characteristics of the quasi-permanent networks described previously, but we did not encounter any publicized long-term (2+ years) stationary deployments of Kunak sensors in our review.


AQMesh is a UK-based sensor package manufacturer which targets many common pollutants: CO, $CO_2$, NO, $NO_2$, $O_3$, $SO_2$, VOC, and PM in addition to less-commonly studied $H_2S$. Data portal and API access are only available to subscribers (AQMesh, accessed 9/18/2023). The AQMesh sensor packages have been used in a wide array of campaign-based research studies (Castell et. al., 2018; Jiao et. al., 2015; Rodríguez et. al., 2020). One study reported in 2017 that the sensors were not
yet ready for research-grade applications (Castell et. al., 2017), zeroing in on the importance of proper calibration.





Ellona's WT1 electronic nose platform is one of the few sensor nodes studied here which is customizable from a long list of sensors, many of which are less frequently studied (including $H_2S$, $NH_3$, HCl, $H_2O_2$, HCHO, $CH_4$, and $H_2$) in addition to the typical pollutant list: CO, $CO_2$, NO, $NO_2$, $O_3$, $SO_2$, VOCs, and PM. Similar to the other campaign-based sensor companies, its

data processing, calibration, and visualization procedures are hidden behind a paywall for users only (Ellona, accessed 9/18/2023). At least one peer-reviewed study utilized the WT1 sensors for campaign-style research efforts as of September 2023 (Panzitta et. al., 2022). Ellona is based in France, and their website boasts usage in countries that appeared infrequently elsewhere in our review, including Estonia and Canada (Ellona Customer Cases, accessed 2/12/2024).

India-based Prana Air builds sensor packages offering a similar sensor range: CO, $CO_2$, NO, $NO_2$, $O_3$, $SO_2$, VOC, $H_2S$, $NH_3$, and PM. Sensor packages can be added to their AQI mobile app, which also includes data from regulatory monitoring stations. According to their website, the sensor packages are calibrated "distinctively" (Prana Air, accessed 9/18/2023). Studies using the Prana Air monitors seem to focus on PM (Patwardhan et. al., 2021; Kumar and Doss, 2023).

Oizom is another privately-owned sensor package company with offices in India and France. Their "Polludone" platform measures CO, $CO_2$, VOCs, and PM. The sensors include unspecified built-in calibrations and a software package, including live data monitoring and API options. Oizom sensors appear to be gaining international traction, with their website boasting usage of their products in more than 50 countries (Oizom, accessed 2/7/2023); peer-reviewed outdoor applications include traffic monitoring in Germany (Kurtenbach et. al., 2022) and emissions from fireworks in India (Shankar et. al., 2023).


Serbian company ekoNet manufactures sensors that measure CO, $CO_2$, NO, $NO_2$, $O_3$, $SO_2$, and PM. Like most of the companies in this category, web and app-based data visualization is limited to subscribers, with three different packages available: basic, standard, and advanced (ekoNet, accessed 9/18/2023). The ekoNet sensors have been utilized in several peer-reviewed publications to date (Vajs et. al., 2021; Jovašević-Stojanović et. al., 2015; Drajic et. al., 2020).

**3.3 Campaign, operated by government**

The two examples of campaign-style low-cost sensor deployments organized by government agencies listed in this section both consist of in-house built sensor packages deployed for the governing body to measure key pollutants in certain regions as well as to better understand the use of low-cost sensors themselves. These reports include calibration and data validation efforts such that the governing agency learned best practices to pass on to their constituents.


The US EPA has sponsored various low-cost sensor projects over the years. Its Village Green project focused on community outreach using low-cost sensor packages measuring $O_3$ and PM, later expanding to $NO_2$, VOCs, and black carbon. The project deployed these sensor packages campaign-style in a handful of US cities, including: Durham, North Carolina; Houston, Texas;





Washington, DC; Kansas City, KS; Philadelphia, PA; Oklahoma City, OK; Hartford, CT; and Chicago, IL (US EPA Village
Green Project, accessed 9/18/2023). Sensor data was compared with that of nearby regulatory grade monitors to ensure data
quality (Jiao et. al., 2015).

The Italian government's ENEA (translated, Italian National Agency for New Technologies, Energy and Sustainable Economic
Development) has likewise sponsored various network sensor efforts in recent years. During the RES-NOVAE (translated,
Networks Buildings Streets: New Challenging Objectives for Environment and Energy) National Project, sensor packages
designed and built by ENEA called AIRBOX were deployed in Bari, Italy to measure: CO, $CO_2$, $NO_2$, $O_3$, $SO_2$, VOCs, and
PM, focusing on roadside emissions (Penza et. al., 2017). Early results comparing the sensors with nearby reference monitors
were promising (Suriano et. al., 2016). ENEA's AIRBOX sensors have been used in a small number of other deployments
(Borrego et. al., 2016). Another ENEA-built sensor package, NASUS, was designed to measure CO, $NO_2$, $SO_2$, $H_2S$, and PM
in outdoor deployments, and proved effective in lab calibrations (Suriano et. al., 2014), outdoor colocations with reference
instruments (Penza et. al., 2014), and during a deployment near a landfill (Penza et. al., 2015).

## 4 Trends across Stationary Low-Cost Sensor Networks

The most commonly occurring gas-phase compounds to appear in our review of stationary gas-phase monitoring networks
varied by deployment type. Even though we selected for gas-phase networks only, networks that also included PM dominated
the quasi-permanent pollutant counts and were sizable in campaign-style deployments as well. $NO_2$, $O_3$, and CO were the most
measured gas-phase pollutants for campaign networks. For quasi-permanent networks, $O_3$ was measured more than any other
gas-phase compound, followed by $CO_2$ and VOCs. Lumping both total and speciated VOCs into one category brings the
number of VOC observations close to those of $CO_2$ in the quasi-permanent category, but the low numbers of VOCs in each
study type highlights the need for more observations of these compounds, which are important in both human health and
climate change contexts.



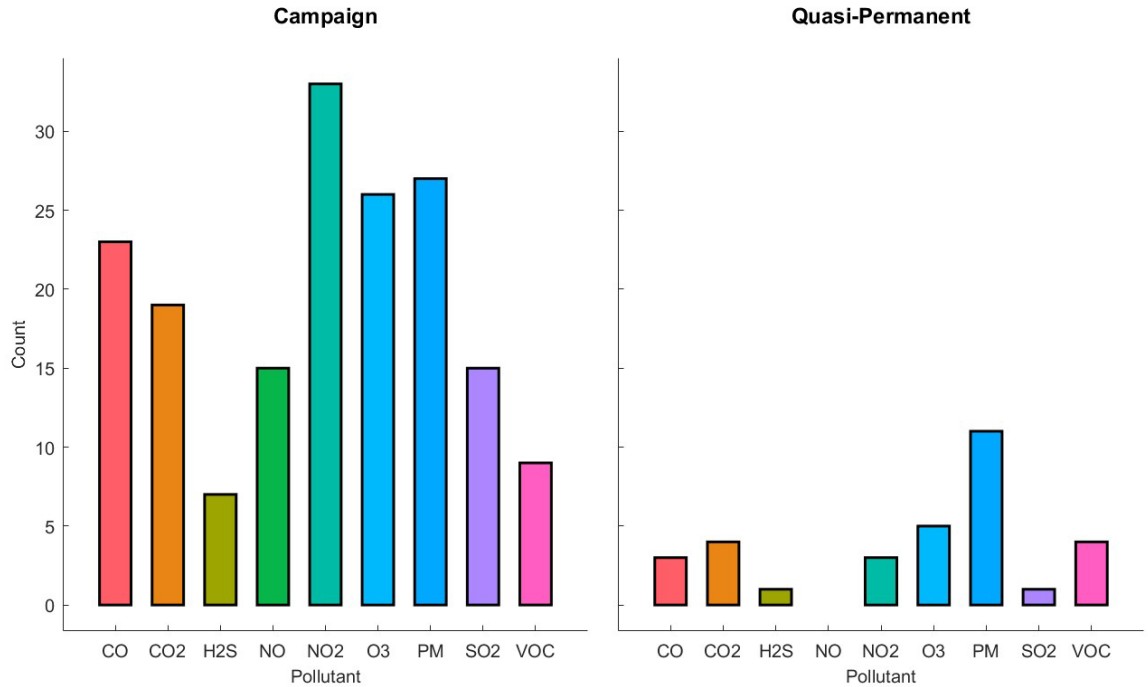

**Figure 3: Histogram of pollutants in each selected sensor package. Note that a sensor package must measure at least one gas-phase compound to be included; no standalone PM sensors are listed here.**

We also sought to characterize global coverage and biases in low-cost sensor placement by mapping the locations of each of the sensor networks studied. Note that for the campaign-based studies, we sought to include at least two networks from each geographic area, although some regions (North America and Europe) were much easier to find information on than others (the Middle East, Central and South America, Oceania), which is reflected in Fig. 4. In the regions with sparser sensor sampling, the networks also tended to be smaller in scale with less longevity compared to more commonly sampled regions; we

encountered networks with few nodes, and more standalone studies as opposed to multiple deployments of the same sensor packages.

   For the quasi-permanent networks, the vast majority were either US-based or featured global coverage, although most of the networks with global coverage had their headquarters based in the US. Overall, we found that monitors are concentrated in the

northern hemisphere mid-latitudes, in high and some middle-income countries, in urban areas. Ironically, low-cost sensors are an attractive solution to this bias in monitoring locations, but clearly, more needs to be done, perhaps by local governments or even sensor companies themselves, to get the technology into less commonly studied areas.



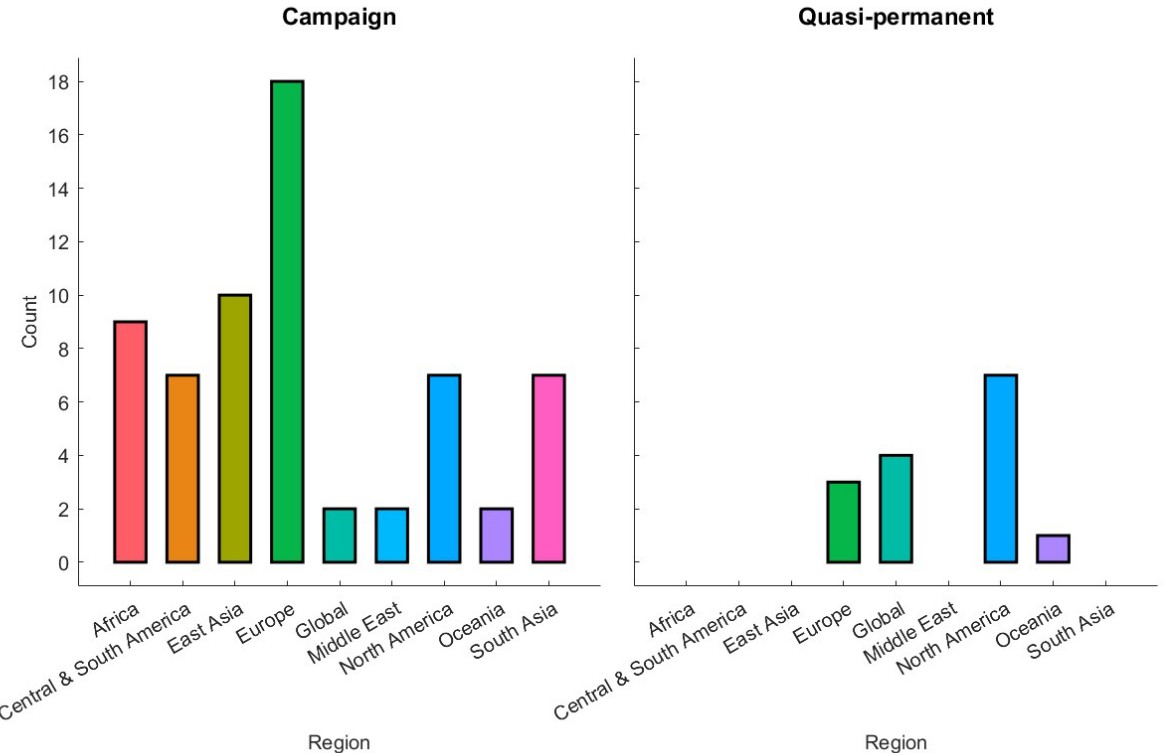

**Figure 4: Histogram of locations of sensor studies, split into broad geographic regions.**

Since one of the biggest issues plaguing low-cost sensor usage is data quality, we also sorted each of the networks we reviewed by their calibration method, as shown in Fig. 5. Most sensor networks reviewed relied on colocation of the sensors with a reference-grade instrument, making this the gold standard for sensor calibration. However, it is important to note that this category is extremely broad, and that variations in the implementation of colocations can still affect sensor data quality considerably.


For instance, some networks may only colocate one of their sensors with the reference instrument (e.g. Love my Air Denver), which does not consider variations among individual sensors. Others that colocate their sensors with a reference instrument may do so as more of a quality check than a true calibration; a sensor is considered "good enough" if concentrations between sensor and reference instrument do not deviate beyond a set range. The most robust in this category are sensor networks that

use this colocation period to derive calibration algorithms, ensuring that the sensor data are as close to the reference data as possible. Even within the algorithm category, some networks may use a single correction factor, while others will use machine learning or other more sophisticated methods to achieve the best fit possible (e.g. Y-pods).





There is also a time component to sensor calibrations due to both sensor age and seasonality. Sensor performance tends to decline as they age (Rai et. al., 2017), with most sensors' useful life hovering around approximately two years (Okorn et. al.,

2021). Beyond this point, their performance may become less reliable, and drift may become evident in the signals. Likewise, low-cost sensors can be very sensitive to environmental conditions, including temperature, pressure, and humidity (Okorn et. al., 2021), so a calibration conducted in hot, humid summer weather may not translate well to cold, dry winter conditions experienced at the same location during different times of year. Thus, these broad-brush categories do not control for the frequency of calibration, nor the robustness of the comparison or algorithm.

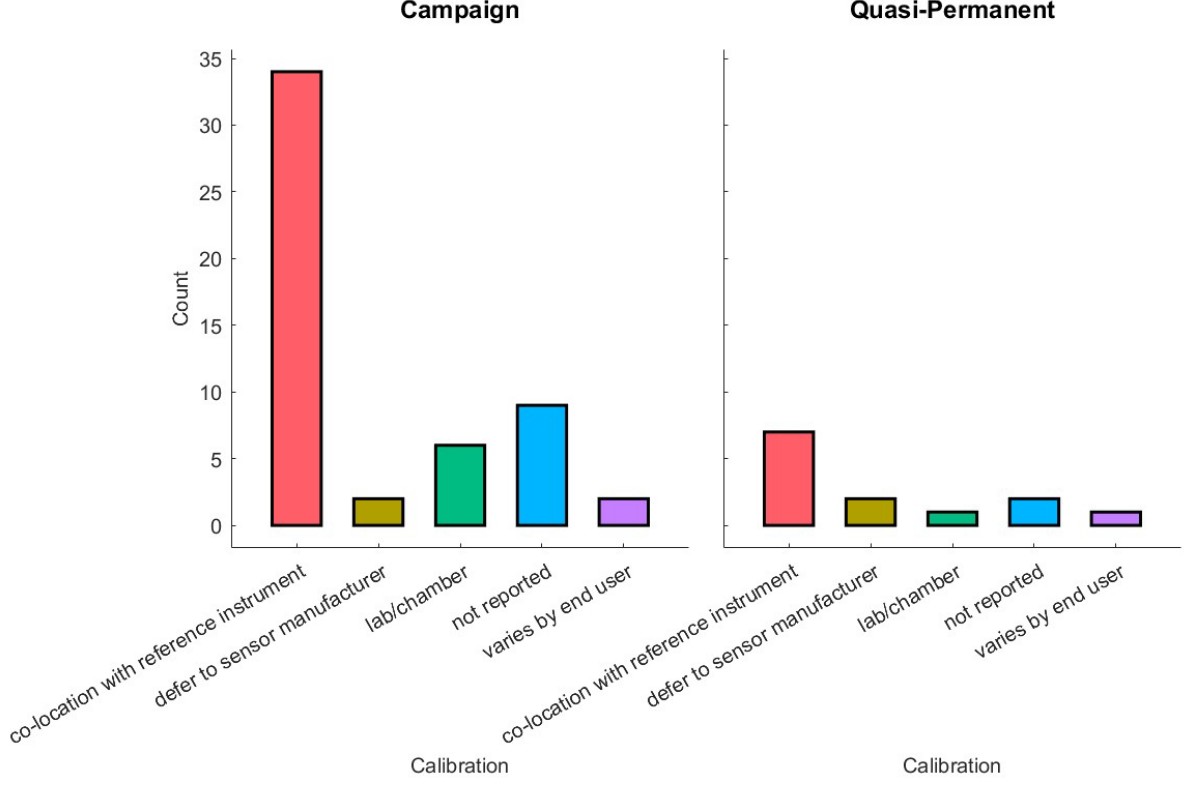


**Figure 5: Histogram of sensor network calibration methods.**

The networks included in the "not reported" calibration category did not provide specific information on their calibration techniques in publications or on their websites at the time of this review. Note that the researchers may have chosen to withhold details for proprietary reasons, in the case of private companies, or may not have performed any calibration on their sensors.

In either case, we did not have sufficient information to categorize their techniques into any other category. The "varies by end user" category implies just that: we found several groups employing the sensors in both campaign and quasi-permanent structures, with each group choosing their own calibration scheme for the same sensor. Likewise, several groups in each category pointed back to materials on the sensor's effectiveness as reported by the manufacturer in lieu of conducting their



own calibration. Again, due to the environmental dependence of low-cost sensors, outdoor colocations are crucial for
understanding how the sensor will respond in its deployment environment, which all other calibration methods are deficient
in.

On a related note, data quality does not necessitate data availability. The highest quality sensor data, calibrated via colocation
with subsequent fitting algorithms, are most likely to hail from university or research groups, which are also the least likely to
ensure public data access. While most data collected by these groups is available upon request, end users outside the scientific
community are less likely to know where to look for data that is not available in a public repository.

## 5 Sensor Network Unification Efforts

In the wake of the abundance of sensor networks springing up worldwide, several websites and databases have emerged in
attempts to highlight noteworthy efforts in the space and make it easier for sensor users and potential data end users to find
sensor networks and data that matches their needs. Most of the sensor unification efforts we studied included lists of sensor
networks with links to their individual websites. Several others include an interface for data visualization – typically a map
with pins for each sensor node, with different colors corresponding to different concentration ranges. These may also include
a data download or API option, but none of the sensor network unification efforts we studied include specifics on data quality
broken down by each sensor type.

As discussed in the previous sections, calibrations are not created equal among sensor networks; while some include vigorous,
frequent calibration procedures, others are seldom calibrated, if at all. Sensor synthesis efforts could be made stronger with
more emphasis on data quality, ensuring that the end user understands the uncertainty of the reported values. Furthermore,
having a standard data format and download option would make it easier for users outside the scientific community to access,
combine, and analyze data that might be relevant to them. Below, we describe several examples of data unification projects,
highlighting both their strengths and limitations.

Several of the long-term monitoring networks discussed in sect. 2 operate de facto data unification products, with their data
maps including data from multiple low-cost sensor networks and regulatory monitors (e.g. AirNow's Fire & Smoke Map,
Clarity, Tellus). These data unification efforts to date have broadly overlooked the low-cost sensor networks established by
university and other small-scale research groups, focusing instead on other companies or non-profit organizations. However,
it is precisely this smaller scale that allows sensor researchers in this space to develop state-of-the-art calibration procedures;
academic institutions tend to have the highest degree of certainty and calibration in their data (see sects. 2.3 and 3.1).



Beyond these, there are a small number of projects that aim to highlight data from a variety of different sensor networks. The
Clean Air Monitoring and Solutions Network (CAMS-Net) is an NSF-funded project which aims to create a "network of
networks", making air quality data from a variety of sources publicly accessible in one digital space, with a focus on low-cost
sensors. As of 2023, their website hosts an extensive list of participating networks, but does not yet have a large, publicly
available data repository (CAMS-Net, accessed 9/18/2023). OpenAQ is a non-profit organization that similarly aims to unify

air quality sensor data, including an interactive map and API for data downloads (OpenAQ, accessed 9/18/2023). RIVM, a
government division in the Netherlands, has created a data portal dubbed "Measuring Together" where citizen scientists can
upload their low-cost sensor data (Wesseling et. al., 2019).

Another major player in the sensor network unification space is the World Air Quality Index, which operates as a non-profit
organization (Real-time Air Quality Sensor Networks, accessed 1/11/2024). The site's strengths include live mapping, basic
plotting, a link to the network's own website, and an option to download historical data. For users wanting to add their own
data to the site, it encourages the use of GAIA sensors, which are PM focused, with the option to add a $CO_2$ sensor only for an
additional cost. However, it also supports data from other sensor manufacturers and networks. Like the other sensor unification
projects, end users wanting more detailed information about each network will need to follow links, which sometimes direct

the user to information on the deployment location rather than the sensor network itself. For instance, a sensor deployed on
the roof of an embassy might link to the embassy's website regardless of whether the embassy or a third party was responsible
for the deployment and upkeep of the sensor, leaving the user with little additional information. Likewise, multiple different
sensor networks may be represented on an individual sub-map, which can make determining the source of data difficult. For
users who are merely interested in the data at face value or want to understand how many monitors are already present in a

certain area, tools like this are immensely helpful in mapping current sensor efforts. However, more advanced end users who
want to understand the measurement and calibration techniques behind each sensor should expect to do a lot more digging to
find this information.

We categorize these projects combining sensor data from a variety of sources in a list or map format as sensor unification
focused, making it possible for end users to locate real-time or historical data that may be personally relevant. While these
projects can be useful, sensor harmonization should be the next step in increasing the usefulness of sensor data from a variety
of networks. Data "harmonization" as used here follows the intent of  Zeb et. al. (2021) of "reconciling various types, levels
and sources of data in formats that are compatible and comparable, and thus useful for better decision-making" and the
principles of Cheng et. al. (2024). In the context of low-cost sensors, harmonization includes taking additional steps to ensure

the seamless use of data: simple enough for citizen scientists to access it, yet quality-assured for scientists and other advanced
users. This is no small task, but steps in this direction might include a common download format for sensor data, availability
of both live and historical data, and information on data quality or calibration procedures. Sensor data harmonization projects
could take sensor unification efforts to the next level by making sensor data more practical for scientific use.



## 6 Emerging and Future Sensor Directions

In the most recent low-cost sensor studies (spanning from approximately 2020 to present), an emerging trend is incorporating sensors into larger scale modeling. Many of the application-based projects included in our review tend to view each sensor data point as individual, e.g. for discerning air quality at a specific point in space and time, typically for comparative purposes with other sensor nodes. In more recent studies, however, we encountered many more applications into geospatial interpolation models such as kriging across sensor nodes to estimate pollutant concentrations not only at sensor locations, but between them
in an even finer grid (Lee et. al., 2024; Liu et. al., 2022; Nori-Sarma et. al., 2020).

While low-cost sensors have long been implemented as a solution to gaps in traditional air quality monitoring techniques, only more recently have they been used to complement other monitoring efforts via data fusion techniques. In these kinds of studies, low-cost sensor measurements are typically coupled with a combination of regulatory monitoring, satellite data, and model
outputs, with different weights assigned to each data stream according to the estimated data quality (Yoo et. al., 2020; Bobbia et. al., 2022; Gressent et. al., 2020; Liang et. al., 2023; Li et. al., 2020). Including low-cost sensor data in these models can improve forecasting (Malings et. al., 2021).

A smaller number of studies we found noteworthy utilize low-cost sensors to estimate pollutant flux across a region, since
finely spaced sensors can essentially act as a grid to estimate across (Turner et. al., 2020). Low-cost gas sensors have also been used in flux chambers to aid evaluation efforts (Bastviken et. al., 2020). Overall, we expect to see the point-source and general monitoring efforts described in sects. 2 and 3 continue to flourish in the next decade. Further evolution of these approaches and advanced modeling-based studies will require improvement in data quality and spatial coverage of trace gas sensing networks.

## 7 Conclusions

Particulate matter is the most frequently studied pollutant using low-cost sensors. In regions with sparse gas-phase sensor data, we could always find several examples reporting particulate matter. Even among the gas-phase networks studied here, most multi-pollutant sensor packages also included a PM sensor, especially among the largest networks. Despite the surge of low-cost sensor networks in the past decade, more monitoring is needed to better address air quality concerns and disparities on
global and regional scales. At present, Central and South America, the Middle East, and Africa are especially lacking. Although larger networks such as PurpleAir service these areas, coverage is significantly less dense compared to the US and western Europe. Even in countries with higher spatial coverage of sensors, rural areas lack the same coverage as their urban counterparts, and countries with higher gross domestic product tend to have more sensors – the differences between western and eastern Europe are a prime example. Less wealthy and more rural areas tend to have more regulatory monitoring gaps;
closing the gap with well-calibrated low-cost sensors is an important first step toward adequate monitoring in underserved

areas. Likewise, satellite and model-derived projections of climate-related impacts will be improved by well-calibrated, globally distributed, high-density surface observations of trace gases, particularly in regions with regulatory monitoring gaps. Forward progress on complicated issues of human health and environmental justice will require significant expansion of the spatial coverage of hydrocarbon trace gas sensors, particularly with attention to detection of specific toxic compounds.


Data quality and availability also plague the use of low-cost sensors. This review found that the highest quality sensor data is obtained by utilizing an outdoor colocation with a reference instrument and subsequent algorithm development to ensure a proper fit. While most campaign and quasi-permanent sensor networks alike rely on a colocation approach, university-based and other research groups typically remain the highest quality. One reason for this is their "gold-standard" post colocation

algorithms, which bring the response of the sensors closer to the true value observed during the colocation period. Another strength of university-based projects is the smaller number of nodes usually deployed; the larger the network, and the more area covered, the less likely that every node has been properly calibrated.

However, data from research-focused networks is typically the hardest to access, with very few publicly available data portals.

Most is available by request only and lacks a standard format, making it inaccessible for most citizen science or education outreach purposes. In contrast, many privately owned sensor companies, particularly in the quasi-permanent category, make their dataset publicly available, despite lacking these rigorous calibration procedures. Research-grade end users have largely deemed global or country-wide calibrations inadequate, especially during wildfires or other extreme weather events. Addressing sensor deployment gaps in trace gases and global spatial coverage, and sensor data gaps in quality and availability

will ensure that low-cost sensors remain a valuable tool for decades to come.

**Code Availability**

The figures in this manuscript were created using the gramm package for MATLAB (Morel, 2018); contact the corresponding author for access.

**Data Availability**

Selected metrics on each of the sensor networks reviewed are included in the supplemental. For additional metrics, contact the corresponding author.

**Author Contributions**

Kristen Okorn: Conceptualization; data curation; formal analysis; writing – original draft preparation



Laura T. Iraci: Conceptualization; writing – review & editing

**Competing Interests**

The authors declare that they have no conflict of interest.

**Acknowledgements**

K.O was supported by the NASA postdoctoral program and the NASA Early Career Investigator Program in Earth Science (23-ECIPES23-0202). L.T.I acknowledges support from the NASA Earth Science Research and Analysis Program.

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
