# Peer review of "An overview of outdoor low-cost gas-phase air quality sensor deployments: current efforts, trends, and limitations"

_EGUsphere, 2024_

## Author Response (AR1)

**Reviewer responses for:** An overview of outdoor low-cost gas-phase air quality sensor deployments: current efforts, trends, and limitations

Kristen Okorn[1,2,3], Laura T. Iraci[1]

[1]Atmospheric Science Branch, NASA Ames Research Center, Moffett Field, CA, 94035, USA
[2]NASA Postdoctoral Program, Oak Ridge Associated Universities, Oak Ridge, TN, 37830, USA
[3]now at Bay Area Environmental Research Institute, Moffett Field, CA, 94035, USA

**Reviewer 1**

Reviewer 1 comment #1: The manuscript devoted to an impressive review on sensor networks and related efforts to monitor air quality worldwide. Significant analysis has been done by deep approach and methodology. Many features have been outlined in different Countries: particulate matter and gases have been reviewed as deployed by universities, research groups and private companies. Some key references should be added to complement the full list.

Author Response #1: Thank you for your thoughtful review of our manuscript. The 2019 Schneider paper is an excellent addition to section 5, where we discuss recent sensor data harmonization efforts. We appreciate your pointing out the newest WMO/IGAC report; it has been cited in section 1.5. Because we were unable to obtain a copy of the book chapter that was mentioned, we do not feel comfortable citing it in any specific way here, but we do appreciate knowing of its existence.

Reviewer 1 comment #2: Minor typo: at page 17, the reference Sun et al., 2016 is related to "Hong Kong Marathon" and not related to "Beijing Marathon". Please, amend properly the text.

Author Response #2: Thank you for catching this error. We have corrected it.

**Reviewer 2**

Reviewer 2 comment #1: It is informative that authors outline the sensor networks according to the organization operating them and the regions of deployment in these two sections. While the readers can obtain plentiful information from the text, I would suggest providing a detailed table in the maintext showing key characteristics of these network, such as the data access, calibration procedure, data quality and better grading the network for scientific researcher.

Author response #1: Thank you for this feedback. We have moved the table with the requested information from the supplemental to the main text of the manuscript (Tables 1&2) and have added some formatting changes for clarity. Regarding which columns are included, we have opted to only include information that is succinct (i.e., does not necessarily require context and caveats for the reader to understand) and unlikely to change. While adding data access

information is a valuable suggestion, networks may publish data only from certain nodes, may technically be available but hard to find, or have inconsistent availability of live and historical data. We feel that this information needs to be distilled with the proper context in the text for each network, and have included as much information as possible on sensors with available data in the main text. As an example, since the publication of our preprint in April, PurpleAir has made significant changes to their API system that would already be out of date and also difficult to encapsulate in a few words in the table. Since the intention of the table is to be quick access information for the reader, we would like to keep it free of additional context and caveats that can be better explained elsewhere.

Reviewer 2 comment #2: Line 495: It seems like most of East Asia deployment of sensor networks authors reviewed were based in China instead of out of China.

Author response #2: Thank you for bringing this to our attention. We meant to state that the majority of these networks were based in China, but see that the verbiage was confusing and have corrected this.

Reviewer 2 comment #3: Suggest adding more information about current status and future trend on the scenarios of sensor deployment, such as UAV and ship.

Author response #3: We appreciate this comment, and have added a paragraph on the use of sensors onboard UAV and recent deployments on ships and in ports to section 6.